# A Possible Antioxidant Role for Vitamin D in Soccer Players: A Retrospective Analysis of Psychophysical Stress Markers in a Professional Team

**DOI:** 10.3390/ijerph17103484

**Published:** 2020-05-16

**Authors:** Davide Ferrari, Giovanni Lombardi, Marta Strollo, Marina Pontillo, Andrea Motta, Massimo Locatelli

**Affiliations:** 1SCVSA Department, University of Parma, 43125 Parma, Italy; 2Laboratory Medicine Service, San Raffaele Hospital, 20132 Milano, Italy; strollo.marta@hsr.it (M.S.); pontillo.marina@hsr.it (M.P.); motta.andrea@hsr.it (A.M.); locatelli.massimo@hsr.it (M.L.); 3Laboratory of Experimental Biochemistry and Molecular Biology, IRCCS Istituto Ortopedico Galeazzi, 20161 Milano, Italy; giovanni.lombardi@grupposandonato.it; 4Department of Athletics, Strength and Conditioning, Poznań University of Physical Education, 61-871 Poznań, Poland

**Keywords:** overtraining, creatine kinase, vitamin D, ROS, testosterone, cortisol, SOD1

## Abstract

The health benefits of physical activity are recognized, however, high levels of exercise may lead to metabolic pathway imbalances that could evolve into pathological conditions like the increased risk of neurological disease observed in professional athletes. We analyzed the plasma/serum levels of 29 athletes from a professional soccer team playing in the Italian first league and tested the levels of psychophysical stress markers (vitamin D, creatine kinase, reactive oxygen species (ROS) and testosterone/cortisol ratio) during a period of 13 months. The testosterone/cortisol ratio was consistent with an appropriate training program. However, most of the athletes showed high levels of creatine kinase and ROS. Despite the large outdoor activity, vitamin D values were often below the sufficiency level and, during the “vitamin D winter”, comparable with those of the general population. Interestingly, high vitamin D values seemed to be associated to low levels of ROS. Based on the results of our study we proposed a vitamin D supplementation as a general practice for people who perform high levels of physical exercise. Beside the known effect on calcium and phosphate homeostasis, vitamin D supplementation should mitigate the high reactivity of ROS which might be correlated to higher risk of neurodegenerative diseases observed in professional athletes.

## 1. Introduction

Physical activity and exercise training are recognized to provide a range of significant benefits associated to both physical and mental health [1,2]. Nevertheless, excessive exercise (EE) may lead to an increased risk of heart dysfunctions as well as altered biological, neurochemical, and hormonal regulation mechanisms [3]. This is common in professional athletes whose training involves an overload period which is often not complemented by an adequate recovery. As a consequence, depending on the time needed to restore performance capacity, athletes may experience overtraining/overreaching syndrome [4] which is frequently associated to catabolic and anabolic imbalance involving skeletal muscle proteins, the neuroendocrine system, and the autonomic nervous system [5,6]. Furthermore, EE has been associated to sport-related skeletal muscle injuries due to repetitive sarcolemma micro-damages and altered calcium homeostasis, bony stress fractures due aberrant loads and accelerated bone turnover, and acute macro-trauma especially in contact sports [7]. If contact sports and musculoskeletal injuries might be seen as an obvious connection of events, a less evident correlation was observed lately between sports and neurodegenerative diseases [8]. Recent studies showed that former professional athletes, who participated in contact sports, had an increased risks of impaired cognitive function and dementia, Parkinson’s disease, Alzheimer disease and amyotrophic lateral sclerosis (ALS) [8,9,10]. However, besides the statistical significance, no clear evidence explaining the association between head trauma and neurodegenerative diseases has been proposed. Furthermore, conflicting results have been published showing an association between high-level chronic physical activity and ALS regardless of any traumatic events [11]. ALS, a fatal adult-onset neurodegenerative disorder often associated with professional sports and soccer in particular [12,13], is characterized by the progressive loss of motor neurons in the brain, brainstem, and spinal cord, which brings paralysis and death within a few years from diagnosis [11,14]. Recent studies showed that the motor neurons impairment in ALS is often associated with the protein misfolding and deposition of superoxide dismutase 1 (SOD1), into insoluble aggregates, likely caused by a structural destabilization induced by gene mutations and/or oxidative damage [15].

Thus, although many studies have shown that “a certain level of exercise is good”, more seems not to be necessarily better and too much might become deleterious [2].

During a season, soccer players face up to several matches in a congested schedule, with as little as 3 days of recovery in between. In this situation a complete recovery is not reached due to the added biochemical stress. Fatigue, indeed, may persist for days after a single match, impairing physical performance and neuromuscular functions, increasing perceptual discomfort (e.g., muscle soreness) and inducing biochemical perturbations (e.g., muscle damage, inflammatory and immunological markers) [16]. For instance, blood creatine kinase (CK) activity remained significantly higher during the 72 h-recovery period [17,18] and the immune function remained altered in 48 h-post match [19]. Accordingly, the accumulation of muscle damage, inflammatory and immune perturbations triggered by consecutive matches and the daily training sessions may hinder the recovery and, consequently, limit the athletes’ readiness and increase the risk of injury [20,21,22].

We retrospectively analyzed the plasma/serum levels of different psychophysical stress markers such as the testosterone (T) to cortisol (C) ratio [23], vitamin D (vitD) [24,25], creatine kinase (CK) [24] and reactive oxygen species (ROS) [26], in an elite soccer team of the Italian first league during an entire season with the aim of determining the seasonal changes and the eventual association with pathological conditions or injuries that emerged during the observation period.

## 2. Materials and Methods

### 2.1. Study Cohort

The entire squad of soccer players, made up by 29 male athletes, aged 18–40 years (25.9 ± 5.0 years), belonging to the A.C. Milan football team of the Italian “Serie A” were included in this retrospective observational study. The averaged heights and weights were, respectively, 183.7 ± 5.9 cm and 77.5 ± 7.7 Kg. The team regularly trained and competed at latitudes with middle/high sun exposure (between 45° and 46° N of latitude), even during autumn and winter. The competitive season started on 20th of August 2017 and ended on 20th of May 2018. Thus, the off-season typically takes place in June and the pre-season starts in July. All individuals involved in the study gave an informed consent to the use of their anonymously collected data for retrospective observational studies (with reference to article 9.2.j of the EU general data protection regulation 2016/679 (GDPR)), according to the San Raffaele Hospital internal policy (IOG075/2016).

No regular supplementation of vitD was followed by the team, however, a sporadic intake of vitamin D by single athletes cannot be excluded.

### 2.2. Samples Collection

Blood drawings were performed on July 5th, 2017; September 13th, 2017; November 29th, 2017; January 15th, 2018; March 6th, 2018; April 24th, 2018; July 10th, 2018 and August 15th, 2018. Blood, obtained by standard venipuncture into BD-SST II Advance tube, 3.5 mL, 13 × 75 mm from BD (Becton, Dickinsonand Company, NJ, USA) [27], according to the most up-to-date pre-analytical warnings, was immediately refrigerated and brought to the laboratory of the San Raffaele Hospital in Milan.

### 2.3. Clinical Data

VitD, CK activity, ROS, and T/C were evaluated in all samples. Measurements were performed, within 4 h from withdrawal, on a Roche COBAS 8000 [28] (Roche, Basel, Switzerland) using electrochemiluminescence immunoassays (25(OH)D, T and C), and spectrophotometric assays (CK activity and ROS). VitD was measured as serum total 25-hydroxyvitamin D (25(OH)D) which is considered the best indicator of vitD status [29]. ROS concentrations were expressed in Carratteli units (Car/U) where one Car/U corresponds to a H_2_O_2_ concentration of 0.08 mg/100 mL. The T/C ratio was calculated by dividing the two hormone levels both expressed in nm/L. All of the instrumentation was routinely checked, each month, by averaging approximately 25–28 measurements (one each working day) of standard solution at low and high concentrations.

### 2.4. Statistical Analysis

Statistical analyses and graphs were performed with the software Sigmaplot (Systat-Software, Inc. San Jose, CA, USA) and GraphPad Prism v6.01 (GraphPad Software Inc., La Jolla, CA, USA). A linear regression analysis was performed for the whole dataset and for the dataset without the outliers (values which were at least 3 standard deviations away from the mean) to investigate the relationship between each pair of psychophysical markers. Averaged values and their corresponding standard deviation intervals (STD) were also calculated at each withdrawal date.

## 3. Results

### 3.1. Vitamin D

The averaged circannual 25(OH)D variations for the 29 players were compared with that of the general population living at the same latitude [30] Figure 1, panel (A). During the “vitD winter” (November to March) the athlete group showed average 25(OH)D levels similar to those of the general population Figure 1, panel (A) whereas in the warmer season their averaged 25(OH)D levels were higher and not included in the STD intervals of the general population. Four players (players 15, 16, 20 and 24) were of African origins and showed very low averaged 25(OH)D levels Figure 1, panel (A). The 25(OH)D levels from each player, as well as their averaged values, are shown in the Appendix A. Of the 195 data collected only 41% were above the 30 ng/mL sufficiency threshold [30] whereas 44.6% had insufficient levels of 25(OH)D (20–30 ng/mL) and 14.4% were 25(OH)D deficient (<20 ng/mL) [27]. Most of the 25(OH)D deficient measurements (18 out of 28; 64.3%) were from the four players of African origins which showed levels between 5.7 and 20.4 ng/mL (Appendix A).

As expected, the majority of the 25(OH)D insufficient/deficient levels (<30 ng/mL) were in the “vitamin D winter” (November to March) [30]. During this 5-month interval, 74.6% of the collected blood samples were below the suggested 30 ng/mL limit. In contrast, during the warm season only 41.1% of the samples were 25(OH)D insufficient/deficient.

### 3.2. CK

The averaged CK activities were above the normal clinical range (20–195 U/L) during the entire period of observation Figure 1, panel (B). Appendix A shows that 74.9% of the samples analyzed exceeded the upper limit. The highest percentage of samples above the 195 U/L limit were observed both in the pre-seasons (between July and August) and at the beginning of the season (September to November), whereas in the central and final parts of the season the percentage of samples above the 195 U/L limit was between 58% and 67% Figure 1, panel (B). Of the 195 samples, 22 were collected when the corresponding athletes were on an injured period (Appendix A) yet, 14 of them (63.6%) had CK activity levels still above the limit (Appendix A).

### 3.3. Reactive Oxygen Species

ROS exceeded the suggested upper limit of 300 Car/U in 42.6% of the measurements (Appendix A). The highest percentages of athletes above the limits were recorded in the central part of the season: November 29th, 2017 (59.2%), January 15th, 2018 (72.4%) and March 6th, 2018 (63.0%) Figure 1, panel (C). Of the 22 blood samples taken from soccer players on an injury period, only 3 (13.6%) exceeded the normal range.

### 3.4. T/C Ratio

The averaged T/C ratios were above the 0.76 level, considered as the threshold below which there is a risk of overtraining [31], for the whole season Figure 1, panel (D). Of the 195 collected samples, 183 (93.8%) were in the normal range whereas only one athlete, at the beginning of the season (July, 5th, 2017), showed a T/C ratio < 0.50 consistent with a high risk of overtraining (Appendix A). Ten samples showed T/C levels between 0.58 and 0.75 consistent with a moderate risk of overtraining and only one sample was between 0.50 and 0.57, thus consistent with a substantial overtraining risk. Of the 22 blood samples taken from soccer players on an injury period, only one (4.5%) was below the 0.76 threshold.

### 3.5. Correlation between Psychophysical Markers

Table 1 shows the parameters obtained from the analysis of the linear correlations between each pair of psychophysical markers Figure 2. Among the six couples analyzed (25(OH)D vs. ROS; 25(OH)D vs. T/C; 25(OH)D vs. CK; ROS vs. T/C; ROS vs. CK; T/C vs. CK) we observed a strong correlation between 25(OH)D and ROS (slope: −1.03 and *P* = 0.003) and a weaker, yet significant, correlation between 25(OH)D and T/C (slope: −0.01 and *P* = 0.009). All of the other combinations showed no significant deviation from horizontal (Table 1). The same analysis was performed without outliers: the results confirmed the previous observation with the exception of the correlation between 25(OH)D and T/C which showed no significant deviation from horizontal (*P* = 0.105, data not shown).

## 4. Discussion

We retrospectively studied the psychophysical markers in elite soccer players during a competitive season in order to verify/recognize whether the large loads of physical activity carried out by professional athletes could raise concerns about their physical health. The T/C ratio has been used as a marker of overtraining [31] based on the assumption that free testosterone is a marker of anabolism while cortisol is indicative of catabolism. Our results showed that although the averaged T/C ratio values (above the 0.76 threshold limit during the whole season) were consistent with an appropriate training program, most of the athletes experienced high levels of CK and ROS. Moreover, despite the significant outdoor activity, during the “vitamin D winter”, vitamin D values were often below the sufficiency level, compared with the general population.

The CK and ROS values were above the normal clinical range limit in 74.9% and 42.6% of the cases, respectively. These percentages became even higher (76.4% and 46.2% for CK and ROS, respectively) if samples taken from athletes on an injury period were omitted.

The CK serum activity, is a physical stress marker widely used in sport [32] but is also a marker of pathological conditions like acute myocardial infarction, myositis and myocarditis, hypothyroidism, myopathies etc. [33]. None of the athletes were affected by any of these conditions, however, CK activity might be above the normal clinical range in healthy subjects as well [34] due to CK leaking into the bloodstream upon muscular injury and, after strenuous physical activity. CK serum activity transiently rises to as much as 30 times the upper limit within 24 to 48 h and then slowly decreases over the next 7 days [35]. Thus, relatively high CK activity levels in professional athletes performing sports involving physical contacts, like soccer, could be considered as normal and are not necessarily associated to an overtraining condition. The highest percentages of samples above the 195 U/L limit, observed in the pre-seasons (between July and August) and at the beginning of the season (September to November), might be consistent with the impact to the newly started training program, after the rest period, and the consequent adaptation. If CK can be considered as a “pure” physical stress marker and does not induce any “side effect” to the athletes, the same is not true for ROS. Reactive oxygen species, also known as free radicals, are formed during the mitochondrial respiration, whose rate is enhanced during exercise, as mitochondrial superoxide or consequently to reperfusion after exercise-induced transient skeletal muscle ischemia [36]. ROS participate in a variety of chemical reactions and are also essential in adaptation to exercise (mitochondrial biogenesis, myofibers regeneration), however, when produced in excess, they can oxidize, and thus damage, a range of biological molecules, including lipids (e.g., membrane phospholipids), nucleic acids (DNA), as well as carbohydrates and proteins [37,38]. High ROS concentrations are associated with a decline in cognitive functions, as observed in some neurodegenerative disorders and age-dependent decay of neuroplasticity [38]. Thus, a long term exposure to this highly reactive species might lead to pathological implications [39]. Interestingly, one of the most common neurodegenerative disease among former professional soccer players is ALS [8] whose etiology is associated with the deposition of SOD1, a metalloenzyme responsible for scavenging free radicals [40], into insoluble aggregates in motor neurons. This amyloid-like formation is probably due to a structural destabilization and/or oxidative damage induced by gene mutations [15]. Because a high level of ROS results in higher transcription and translation of the SOD1 gene [41], we might speculate that an increased ROS level, constantly perpetuated along the athlete’s career, will induce an almost constant SOD1 overexpression and thus promote the concentration-dependent formation of SOD1-amyloid-like aggregates [42] possibly associated with the pathogenesis of this neurodegenerative disease. In our study almost 50% of the measured levels were above the normal clinical limit, posing concerns for the long-term health of the players.

VitD, mainly synthesized by the skin when exposed to ultraviolet B radiation (UVB) [27], refers to a group of related steroid hormones involved in several physiological processes centered on the maintenance of calcium and phosphate homeostasis, as well as iron and zinc [27]. Since UVB is necessary to synthesize the vitD precursor cholecalciferol, vitD deficiency in populations living at high latitudes is common, especially during winter [30]. Although an optimal (25(OH)D) level helps to maintain the musculoskeletal system efficiency [43,44], studies on athletes highlighted a surprisingly high prevalence of vitD insufficiency, both in outdoor and indoor disciplines [45,46].

The average (25(OH)D) of the 29 athletes was, in the cold season, similar to those of the general population living at the same latitude. The four players of African origins do not lower significantly the averaged vitD levels, also, from recent population statistics [47] we expect a similar percentage of people of African origin in the general population living in the Milan area as well. Thus, the comparison between the whole group and the general population can be considered as pertinent. As previously observed in other studies [48] we noticed a scenario of insufficiency whilst deficiency was observed mainly in the four athletes of African origins, primarily due to skin pigmentation [49]. This was quite surprising because professional soccer players spend most of their time outdoors and, at this latitude even in the cold seasons, 2 h of sun exposure with 10% of body exposure at solar noon are sufficient for an optimal vitamin D dose [50]. This might be tentatively explained by the training outfit of the athletes which, in the cold season, usually covers a large part of the body. In contrast, during the warm season, where the training is usually performed with short sleeves and short pants, the athletes group showed averaged vitD levels much higher than the general population living at the same latitude.

Interestingly we found a significant correlation between (25(OH)D) and ROS (Table 1) which might confirm the recently discovered antioxidant role attributed to this hormone [51]. A weak yet significant correlation was found also between (25(OH)D) and the T/C ratio. Higher levels of vitD were thus associated to a slightly increased risk of overtraining. The correlation might indicate a possible biological interaction between the hormones, however, because of the rather high *P* value as well as the lack of correlation when outliers were removed, further studies, involving larger dataset, are needed to investigate this issue.

Although no correlation between vitD status and athletic performance have been shown previously [45], and lower rates of osteoporotic fractures have been recorded in African-Americans having insufficient vitD levels [49], there are now evidences that vitD protects against other chronic conditions, including cardiovascular disease, diabetes, and some cancers [49]. The possible antioxidant effect observed in this study might be a further reason to suggest a vitD supplementation to professional athletes.

## 5. Conclusions

Although modern training programs seem to avoid the risk of overtraining in professional soccer players, others psychophysical stress markers, like the free radicals, are often above the normal clinical limit posing as a long term risk for several pathological situations like neurodegenerative diseases. Our data seems to associate an antioxidant effect with normal/high vitD levels. Thus, in the light of the general vitD insufficiency observed in the 29 professional soccer players, we suggest that vitD supplementation should become a general practice for professional soccer players and athletes that have to cope with high ROS levels for a long period of their life. Supplementation will preserve athletes from the harmful skeletal effects of low level of vitD as well as mitigate the detrimental high reactivity of ROS capable of damaging nucleic acids and protein conformation.

## Figures and Tables

**Figure 1 ijerph-17-03484-f001:**
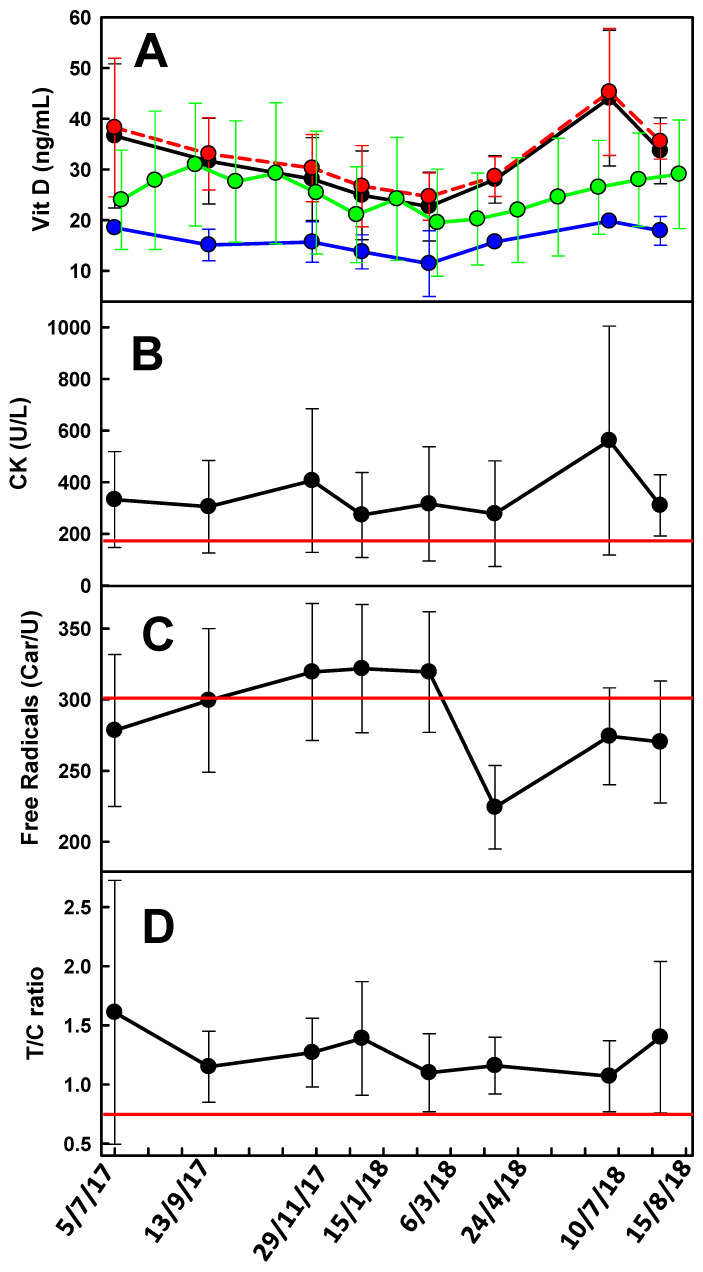
Seasonal variability of the averaged recorded psychophysical stress markers for the 29 soccer players involved in this study. The exact withdrawal date is shown on the x axis. Panel A: Vitamin D (vitD) values for the 29 soccer players (black line), the 4 soccer players of African origins (blue line), and the 25 soccer players of non-African origins (red line). The green line represents the vitD levels of the general population living at the same latitude [30]. Panel B: creatine kinase (CK) values; the red line represents the normal clinical upper limit of 195 U/L. Panel C: free radicals; the red line represents the normal clinical upper limit of 300 Car/U Panel D: testosterone (T)/cortisol (C) ratio; the red line represents the lower limit consistent with an overtraining risk (<0.76). Error bars represent the standard deviation (STD) interval.

**Figure 2 ijerph-17-03484-f002:**
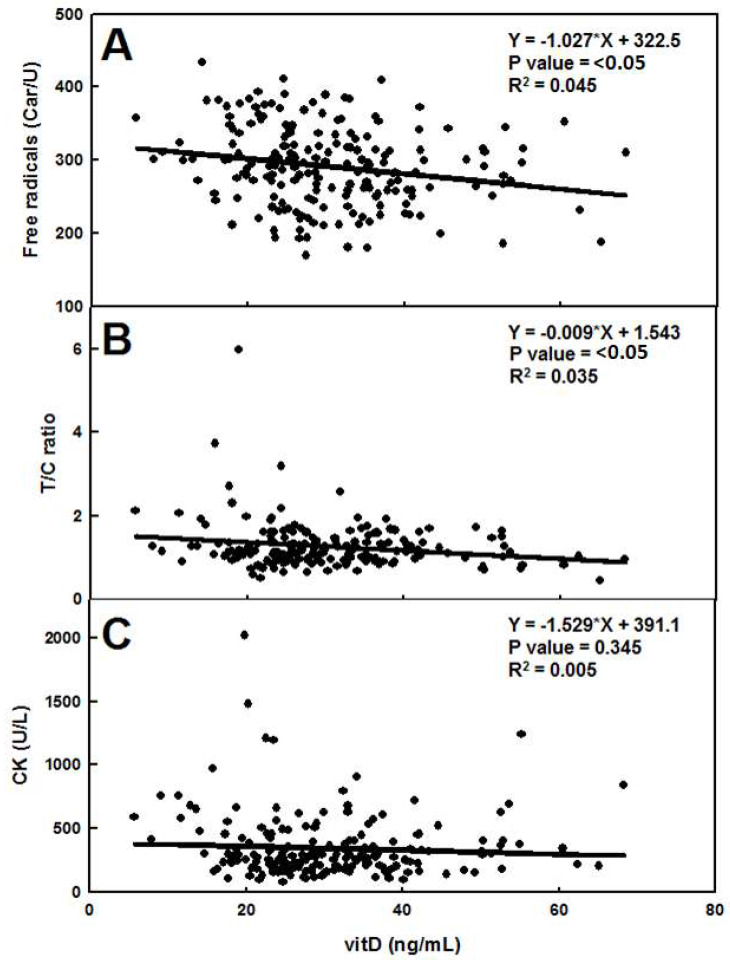
Linear correlation between vitD and free radicals (**A**), vitD and T/C (**B**), vitD and CK (**C**). For each regression the corresponding equation, the R^2^ and the *P* value are shown.

**Table 1 ijerph-17-03484-t001:** Linear regression parameters obtained by comparing VitD, CK and reactive oxygen species (ROS). Deviation from horizontal was considered significant if the *P* value was <0.05.

	Equation	R^2^	P Value	Deviation from Horizontal
VitD vs. ROS	−1.027*X + 322.5	0.045	**<0.05**	**SIGNIFICANT**
VitD vs. CK	−1.529*X + 391.1	0.005	0.345	NOT SIGNIFICANT
VitD vs. T/C	−0.009*X + 1.543	0.035	**<0.05**	**SIGNIFICANT**
CK vs. ROS	0.537*X + 187.6	0.013	0.107	NOT SIGNIFICANT
CK vs. T/C	−0.000*X + 1.317	0.017	0.071	NOT SIGNIFICANT
T/C vs. ROS	0.000*X + 1.213	0.000	0.874	NOT SIGNIFICANT

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
