# Peer review of "A Possible Antioxidant Role for Vitamin D in Soccer Players: A Retrospective Analysis of Psychophysical Stress Markers in a Professional Team"

_ijerph, 2020, doi:10.3390/ijerph17103484_

Round 1

Reviewer 1 Report

Retrospective analysis of psychophysical stress markers in a professional soccer team reveals a possible antioxidant role for vitamin D.

This study presents a very interesting topic based on the elite level in sport. There is a great interest to find strategies to keep our athletes healthy.

This well written manuscript is pleasant to read. Nevertheless, there is a point to check in the results and the authors should show more quickly the main finding of this work in the discussion section.

Introduction

I think the authors can added a sentence about the objective of their study in the end of the introduction.

Materials and method

Please provide more information about players (weight, size, …)

Statistical analysis

  1. 111 – 112 – Please remove one of the words “between”.

Which correlation test did you use?

I do not understand why the graphs are not in the results section. Please move them.
It would be easier to read the graphs if you place the sampling time on the X-axis.

Figure 2 and 3 – Define in the legend what the red line is.

Results

I am not sure that the tables are essentials. We have the average data in the graphs, and these are the most important information. I think you can remove all the tables.

I do not understand why the Figure 4 is not with the three first graphs.

Figure 5 - A very interesting figure. Please use the usual description for the P values (p < 0.05; p < 0.01; p < 0.0001; NS).

Even if these results are very interesting and pleasant, I have a problem with the extreme values in the B and C graphs. What is the origin of these values? Something happened in this player to have these scores. The problem is that these values have a great influence on the statistical results. Did you try to analyze the correlation without these values? Is the P value significant for the correlation between T/C ratio and VitD?

Discussion

Please begin the discussion by your main results. This part arrives too late (third paragraph of the discussion section…).

  1. 252 – Even if I understand what you want to mean, I am not sure that you can begin a sentence with “A weaker, …”

Author Response

We thank the reviewer for his/her valuable suggestion which certainly improved the qulity of the paper.

Changes in the manuscript have been highlighted in red to help the reviewers' work.

Reviewer 1

This study presents a very interesting topic based on the elite level in sport. There is a great interest to find strategies to keep our athletes healthy.

This well written manuscript is pleasant to read. Nevertheless, there is a point to check in the results and the authors should show more quickly the main finding of this work in the discussion section.

Introduction

 I think the authors can added a sentence about the objective of their study in the end of the introduction.

A sentence about the objective of the study has been added at the end of the Introduction section. Also the introduction has been modified according to the other reviewers' request.

Materials and method

Please provide more information about players (weight, size, …)

Height, weight as well as age were included in the materials and methods section.

 Statistical analysis

 111 – 112 – Please remove one of the words “between”.

Thanks for the observation. The typo has been corrected

Which correlation test did you use?

The information was added in the M&M section.

I do not understand why the graphs are not in the results section. Please move them.
It would be easier to read the graphs if you place the sampling time on the X-axis.

Graphs were included in the results section and the sampling time was added on the x-axis.

Figure 2 and 3 – Define in the legend what the red line is.

The red line was defined in the legend of fig 2 and 3 which are now merged in a single figure (Fig. 1) as suggested by the other reviewer.

Results

I am not sure that the tables are essentials. We have the average data in the graphs, and these are the most important information. I think you can remove all the tables.

We removed the tables from the main manuscript and added them as a supplementary file. We hope this change satisfy the reviewer request.

I do not understand why the Figure 4 is not with the three first graphs.

Figure 4 was merged with the previous figures (1, 2 and 3) as suggested by the other reviewers

Figure 5 - A very interesting figure. Please use the usual description for the P values (p < 0.05; p < 0.01; p < 0.0001; NS).

The Figure and table have been modified by introducing the usual description (p<0.05)

Even if these results are very interesting and pleasant, I have a problem with the extreme values in the B and C graphs. What is the origin of these values? Something happened in this player to have these scores. The problem is that these values have a great influence on the statistical results. Did you try to analyze the correlation without these values? Is the P value significant for the correlation between T/C ratio and VitD?

Thanks for the observation. Unfortunately we cannot go back to the origin of the outliers values. However, by analyzing the correlation without them, everything remained unchanged except the vitD vs T/C ratio linear regression where the weak correlation disappeared when outliers were removed. This was included in the text.

Discussion

Please begin the discussion by your main results. This part arrives too late (third paragraph of the discussion section…).

A sentence about the overview of the results have been added in the first paragraph of the discussion section.

252 – Even if I understand what you want to mean, I am not sure that you can begin a sentence with “A weaker, …”

Amended in “A weak yet significant…”

Reviewer 2 Report

The manuscript titled “Retrospective analysis of psychophysical stress 2 markers in a professional soccer team reveals a 3 possible antioxidant role for vitamin D” investigates the expression of psychophysical stress markers (i.e., vitamin D, creatine kinase, ROS and testosterone/cortisol ratio) into the plasma/serum of 29 athletes from a professional soccer team during a period of 13 months. The aim was to verify whether high levels of exercise activity can produce altered expression of these markers and lead to pathological situations. The study revealed that the athletes showed a testosterone/cortisol ratio consistent with an appropriate training program, whereas most of them exhibited high levels of creatine kinase and ROS. Despite the outdoor exercise activity, vitamin D values were often below the sufficiency level and high vitamin D values seemed to be associated to low levels of ROS. Based on this evidence, the Authors proposed a vitamin D supplementation as a general practice for people who perform high levels of physical exercise.

The work is quite interesting and falls within the scopus of International Journal of Environmental Research and Public Health. The Authors make clear the intended practical application of the research, as well as its novelty. Manuscript is well written and English language is appropriate. The conclusions appear to be well supported by the results.

However, there are some problems in the manuscript structure and organization that make it difficult to read. Below are some issues that need to be addressed:

  • The title seems more like a complete sentence. It should be re-written in a more concise and immediate way to give the idea of the investigated topic.
  • The last part of the Introduction section should be revised. The description of the study (page 2, lines 59-65) should not give some details which are more appropriately described in the Material and Methods Moreover, the description of the investigated biomarkers (page 2, lines 66-85) should be moved and integrated to the Discussion section.
  • The Material and Methods section needs to be better organized. It would be preferable to divided it into subsections, corresponding to investigated subjects, sample collection, biomarker evaluation and statistical analysis.
  • Figures 1, 2 and 3 need to be included not before but within the Results It could be more immediate to merge the graphs of Figures 1, 2, 3 and 4 into only one figure.
  • The Result section should be better organized into subsections corresponding to the different investigated markers.
  • Regarding the experimental settings, the Authors should consider to also analyse a control sample (plasma from subjects who do not play high levels of exercise activity) in order to set a reference for basal biomarker expression.

Author Response

We thank the reviewer for his/her valuable suggestion which certainly improved the quality of the paper.

Changes in the manuscript have been highlighted in red to help the reviewers' work.

Reviewer 2

The manuscript titled “Retrospective analysis of psychophysical stress 2 markers in a professional soccer team reveals a 3 possible antioxidant role for vitamin D” investigates the expression of psychophysical stress markers (i.e., vitamin D, creatine kinase, ROS and testosterone/cortisol ratio) into the plasma/serum of 29 athletes from a professional soccer team during a period of 13 months. The aim was to verify whether high levels of exercise activity can produce altered expression of these markers and lead to pathological situations. The study revealed that the athletes showed a testosterone/cortisol ratio consistent with an appropriate training program, whereas most of them exhibited high levels of creatine kinase and ROS. Despite the outdoor exercise activity, vitamin D values were often below the sufficiency level and high vitamin D values seemed to be associated to low levels of ROS. Based on this evidence, the Authors proposed a vitamin D supplementation as a general practice for people who perform high levels of physical exercise.

The work is quite interesting and falls within the scopus of International Journal of Environmental Research and Public Health. The Authors make clear the intended practical application of the research, as well as its novelty. Manuscript is well written and English language is appropriate. The conclusions appear to be well supported by the results.

The authors are grateful to the Reviewer about the positive comment and have taken into account all the constructive comments and suggestions that this Reviewer has made.

However, there are some problems in the manuscript structure and organization that make it difficult to read. Below are some issues that need to be addressed:

The title seems more like a complete sentence. It should be re-written in a more concise and immediate way to give the idea of the investigated topic.

A new version of the title has been proposed: “A possible antioxidant role for vitamin D in soccer players:  a. retrospective analysis of psychophysical stress markers in a professional team”

The last part of the Introduction section should be revised. The description of the study (page 2, lines 59-65) should not give some details which are more appropriately described in the Material and Methods Moreover, the description of the investigated biomarkers (page 2, lines 66-85) should be moved and integrated to the Discussion section.

The introduction section has been thoroughly revised according to this Reviewer’s suggestion.

The Material and Methods section needs to be better organized. It would be preferable to divided it into subsections, corresponding to investigated subjects, sample collection, biomarker evaluation and statistical analysis.

The Materials & Methods section has been reorganized according to this Reviewer’s suggestion

Figures 1, 2 and 3 need to be included not before but within the Results It could be more immediate to merge the graphs of Figures 1, 2, 3 and 4 into only one figure.

Figures were merged into a single figure (figure 1) which was also included in the results section and not in material and methods.

The Result section should be better organized into subsections corresponding to the different investigated markers.

Subsections have been added in the results section

Regarding the experimental settings, the Authors should consider to also analyse a control sample (plasma from subjects who do not play high levels of exercise activity) in order to set a reference for basal biomarker expression.

The control group suggested by the reviewer is an interesting idea. Unfortunately it is not feasible in a short time. However, we adopted a similar approach by using the normal clinical range limits (see red lines in Fig. 1) which are calculated on large groups of “normal” subjects who do not play high levels of exercise activity. Thus it should set as an appropriate reference for basal biomarker expression.

Round 2

Reviewer 1 Report

The modification has improved the quality of manuscript.

Reviewer 2 Report

The Authors responded appropriately to the Rreviewer's suggestions, improving the quality of the manuscript. The work can be accepted in its revised form.